# CCT6A knockdown suppresses osteosarcoma cell growth and Akt pathway activation *in vitro*

**Weiquan Zeng**[1,2,3☯], **Meizhu Wu**[2,3☯], **Ying Cheng**[2,3☯], **Liya Liu**[2,3], **Yuying Han**[2,3], **Qiurong Xie**[2,3], **Jiapeng Li**[4], **Lihui Wei**[2,3], **Yi Fang**[2,3], **Youqin Chen**[5], **Jun Peng**[2,3]*, **Aling Shen** [2,3]*

**1** Department of Orthopaedics, Affiliated Rehabilitation Hospital of Fujian University of Traditional Chinese Medicine, Fuzhou, Fujian, China, **2** Academy of Integrative Medicine, Fujian University of Traditional Chinese Medicine, Fuzhou, Fujian, China, **3** Fujian Key Laboratory of Integrative Medicine on Geriatrics, Fujian University of Traditional Chinese Medicine, Fuzhou, Fujian, China, **4** Department of Physical Education, Fujian University of Traditional Chinese Medicine, Fuzhou, Fujian, China, **5** Department of Pediatrics, Case Western Reserve University School of Medicine, Rainbow Babies and Children's Hospital, Cleveland, Ohio, United States of America

☯ These authors contributed equally to this work.
* pjunlab@hotmail.com (JP); saling86@hotmail.com (AS)

**Data Availability Statement:** All relevant data are within the paper and its Supporting information files.

**Funding:** National Natural Science Foundation of China (81803882). Funders:AS. He is

## Abstract

We assessed the role of the protein-coding gene chaperonin-containing TCP1 subunit 6A (CCT6A) in osteosarcoma, as this is currently unknown. Using data from the R2 online genomic analysis and visualization application, we found that CCT6A messenger ribonucleic acid (RNA) expression is increased in osteosarcoma tissue and cells. Transfection of CCT6A small interfering RNA into cultured osteosarcoma cells revealed that CCT6A knockdown attenuates cell growth, cell viability, cell survival, and induced apoptosis and cell cycle progression at the G0/G1 phases. Moreover, CCT6A knockdown downregulated phosphoprotein kinase B (p-Akt), cyclinD1 and B-cell lymphoma-2, whereas upregulated Bcl-2-associated X-protein expression. Thus, CCT6A knockdown inhibits cell proliferation, induces cell apoptosis, and suppresses the Akt pathway.

## Introduction

Osteosarcoma (OS) is an aggressive form of bone tumor and the third leading cause of cancer-related death in children and adolescents [1, 2]. However, surgical resection combined with the development of effective neoadjuvant and adjuvant chemotherapies has improved prognosis and long-term survival over the last few decades [3, 4]. Yet, due to chemotherapy resistance and metastasis, the overall prognosis for OS remains unsatisfactory, with over 40% of OS patients suffering recurrence or progression of the disease after first-line therapy [5, 6]. Therefore, the elucidation of the molecular mechanisms involved and the development of novel diagnostic and therapeutic strategies are required.

Chaperonin-containing tailless complex polypeptide 1 (CCT) is a molecular chaperone required in the folding of the major cytoskeletal proteins actin and tubulin [7]. Cancer studies have demonstrated that CCT participates in cell cycle progression [8], tumor growth [9–11], and migration [12, 13] of cancer cells. However, the roles and precise mechanisms of CCT

Corresponding author. https://www.nsfc.gov.cn. Natural Science Foundation of Fujian Province (2020J06026). Funders:AS. He is Corresponding author.

**Competing interests:** The authors have declared that no competing interests exist.

**Abbreviations:** CCK-8, cell counting kit-8; CCT, chaperonin-containing tailless complex polyeptide 1; CCT6A, chaperonin containing TCP1 subunit 6A; ECL, chemiluminescence; FBS, fetal bovine serum.

subunits in cancer remain largely unknown. Chaperonin-containing TCP1 subunit 6A (CCT6A) is a zeta chaperonin subunit with two identical rings stacked back-to-back [14–16]. It is a subunit of the CCT family, involved in the correct folding and oligomerization or polymerization of native proteins, and therefore, participates in cell cycle progression and cytoskeletal organization [17–19]. CCT6A has also been reported to be involved in the phosphorylation of extracellular signal-regulated kinase (ERK) [20–22].

Previous research on cancer has found that CCT6A expression is upregulated in drug-resistant human melanoma cell lines [23, 24]. Functional studies in lung cancer have demonstrated that CCT6A overexpression increases the percentage of formed sphere-forming and side population cells and decreases the sensitivity of A549 cells to anoikis treatment. CCT6A has also been shown to suppress decapentaplegic homolog 2 (SMAD2) function in non-small cell lung cancer (NSCLC) cells and to promote metastasis by directly binding to SMAD2 protein, while the selective inhibition of CCT6A efficiently suppresses TGF-β-mediated metastasis [25]. However, the role of CCT6A in most cancers, including OS, remains unknown.

To explore the role of CCT6A in OS, we analyzed the expression of CCT6A messenger ribonucleic acid (mRNA) in OS tissues and cell lines using the R2 online genomics analysis and visualization application [26–28]. We also assessed the role of CCT6A in cell proliferation and apoptosis in OS cell lines after transfection with CCT6A small interfering RNA (siRNA), by observation of cell confluence and colony formation, cell counting, cell-counting kit (CCK-8) assay, propidium iodide (PI) staining, and annexin V staining. The underlying mechanism of CCT6A knockdown in cell growth was explored by determining the expression of related proteins.

## Material and methods

### Materials

McCoy's 5a medium, fetal bovine serum (FBS), a cell cycle kit, a Lipofectamine RNAiMAX transfection reagent bicinchoninic acid (BCA) protein assay kit and chemiluminescence (ECL) detection kit were all purchased from Thermo Fisher Scientific (Waltham, MA USA). A mixture of penicillin and streptomycin was obtained from Hyclone (Logan, UT, USA). An annexin V staining kit was provided by AAT Bioquest (Sunnyvale, CA, USA). CCT6A antibody was provided by Sangon Biotech (Shanghai, China). Antibodies for protein kinase B (Akt), phospho-Akt (p-Akt), B-cell lymphoma-2 (Bcl-2), Bcl-2-associated X-protein (Bax), cyclinD1, cyclin-dependent kinase 4 (CDK4), glyceraldehyde 3-phosphate dehydrogenase (GAPDH), and horseradish peroxidase (HRP)-conjugated secondary antibodies were obtained from Cell Signaling Technology Inc (Beverly, MA, USA).

### Bioinformatics analysis

The mRNA level of CCT6A in OS tissue (GEO ID: gse14359) and cells (GEO ID: gse11414) were analyzed using the R2 Genomics Analysis and Visualization Platform (http://r2.amc.nl), which is a web-based genomics analysis and visualization application widely used to analyze the expression of genes in diseases, including cancer [26–28].

### Cell culture and transfection

Human OS U-2 cells were purchased from the Chinese Academy of Sciences (Shanghai, China) and cultured in McCoy's 5a medium supplemented with 10% FBS and a 100 unit/ml penicillin and 100 μg/ml streptomycin mixture at 37˚C in a humid atmosphere with 5% carbon dioxide ($CO_2$). Cells were sub-cultured at 80%–90% confluence.

To knockdown the expression of CCT6A, three different human CCT6A siRNA (si-CCT6A) oligonucleotides and control siRNA (si-Ctrl) were synthesized by Shanghai Gene-Pharma Co., Ltd. (Shanghai, China) using the following sequences: CCT6A-homo-432 (si-CCT6A-1): 5`-GGG AAA CAC UUA UAG AUG UTT ACA UCU AUA AGU GUU UCC CTT-3,' CCT6A-homo-778 (si-CCT6A-2): 5`-GCU UAA UCA GAG GGC UUG UTT ACA AGC CCU CUG AUU AAG CTT-3,' CCT6A-homo-835 (si-CCT6A-3): 5`-GGG UGG AGG AUG CAU ACA UTT AUG UAU GCA UCC UCC ACC CTT-3,' and NC (si-Ctrl): sense: 5`-UUC UCC GAA CGU GUC ACG UTT-3`, antisense: 5`-ACG UGA CAC GUU CGG AGA ATT-3`. Cells were seeded and cultured overnight, followed by transfection with siRNA or si-Ctrl (50 nM) using Lipofectamine RNAiMAX according to the manufacturer's instructions.

## Cell confluence observation

U-2 OS cells ($0.4 \times 10^5$ cells/ well) were placed in six-well plates and transfected with si-CCT6As or si-Ctrl for 48 h. Cell confluence was observed and photographed using a phase-contrast inverted microscope (Leica Microsystems Ltd., Wetzlar, Germany) at a magnification of ×200.

## Cell number counting

The transfected cells were trypsinized and diluted with equal volumes of fresh medium and cell-containing medium, which was mixed with 0.2% of Trypan blue solution (Sigma Aldrich, MO, USA). The cells were counted using a Countstar Automated Cell Counter (ALIT Life Science Co., Ltd., Shanghai, China) according to the manufacturer's instructions.

## Cell viability analysis

Cell viability was determined using a CCK-8 assay. Briefly, U-2 OS cells ($2 \times 10^3$ cells/ well) were placed in a 96-well plates and cultured overnight. This was followed by transfection of si-CCT6As or si-Ctrl for 24 h, 48 h, or 72 h. After transfection, 10 μL of CCK-8 was added to each well and the cells were incubated at 37°C for an additional 2 h. Absorbance was measured at 450 nm using an Infinite 200 Pro microplate reader (Tecan, Männedorf, Switzerland). The cell viability on day 1 was set as 1, and cell growth was calculated in reference to day 1.

## Cell colony formation analysis

A cell colony formation assay was performed to determine the U-2 OS cell survival rate. After transfection with si-Ctrl or si-CCT6A-3 (50 nM) for 72 h, the cells were collected and diluted in the medium. Then, 500 cells from each group were reseeded into 12-well plates and cultured in a humid atmosphere at 37°C and 5% $CO_2$ for 8–10 days. The medium was changed every 2–3 days. Finally, the cells were washed twice with phosphate-buffered saline (PBS) and fixed with 4% paraformaldehyde for 15 min. They were then stained for 15 min with 0.01% crystal violet stain (Beyotime Biotechnology, Shanghai, China). Images were taken and the number of colonies formed was calculated by normalizing the colonization rate of the control cells to 100%.

## Cell cycle analysis

The transfected cells were harvested and fixed with 70% ice ethanol overnight at 4°C, followed by staining with PI/ ribonuclease (RNase) solution (Thermo Fisher Scientific) for 30 min. The fluorescence signal was detected using the FL2 channel and the proportion of

deoxyribonucleic acid (DNA) in each phase was analyzed using Modft LT software, version 3.0 (Verity Software House, Inc. Topsham, ME, USA).

### Annexin V staining and cell apoptosis analysis

Annexin V staining and flow cytometry (FACS Caliber, Becton-Dickinson, San Jose, CA, USA) were performed to determine cell apoptosis levels. Briefly, the transfected cells were harvested with trypsin without ethylenediaminetetraacetic acid (EDTA) and washed twice with PBS. The collected cells were incubated with APC annexin V solution for 30 min in the dark. The cells were sorted by fluorescence-activated cell sorting (FACS) and the percentage of positive cells was calculated.

### Western blot analysis

After transfection for 72 h, the cells were lysed using ice-cold lysis buffer containing protease and phosphatase inhibitor cocktails for 30 min. The supernatant was collected and centrifuged at 4°C, 14,000 rpm, for 20 min to remove sediment. A BCA assay was performed according to the manufacturer's instructions to detect the total protein concentration. 50 μg of the total protein was loaded and separated by sodium dodecyl-sulfate polyacrylamide gel electrophoresis (SDS-PAGE) and transferred to polyvinylidene fluoride (PVDF) membranes. The bands were blocked with 5% skimmed milk at room temperature for 1 h, incubated with primary antibodies (dilution 1:1000) overnight at 4°C, and then incubated with secondary antibodies (dilution 1:5000). This was followed by enhanced chemiluminescence detection.

### Statistical analysis

All statistical analyses were conducted using SPSS statistical software version 22.0 (IBM Corp., Armonk, NY, USA). Data from three independent experiments were presented as means ± standard deviations. Differences between the two groups were tested using independent student $t$-tests. Those between three or more groups were analyzed using one-way ANOVAs. A $p$-value $<0.05$ was considered statistically significant.

## Results

### CCT6A is highly expressed in osteosarcoma tissue and cells

Using the R2 platform [26–28], we analyzed the mRNA expression of CCT6A in OS tissue and cells. As shown in Fig 1A, there was significantly greater mRNA expression of CCT6A in OS tissue than in non-neoplastic primary osteoblasts ($p <0.05$). This increase of CCT6A in OS tissue was confirmed in a comparison of cultured OS cells (U-2 and MG63 cell lines) with normal human osteoblasts ($p <0.05$). Thus, the upregulation of CCT6A expression was shown to be significantly greater in OS tissue and cultured cells than in non-neoplastic primary osteoblasts, suggesting that CCT6A may play an essential role in the development of OS.

### CCT6A knockdown suppresses U-2 OS cell growth

Given the increase of CCT6A expression in U-2 OS cells, we further analyzed the function of CCT6A by transfection of three different types of CCT6A-specific siRNA into U-2 OS cells. Western blot analysis demonstrated that the transfection of both si-CCT6A-2 and si-CCT6A-3 decreased the expression of CCT6A proteins (Fig 2A). Moreover, the transfection of si-CCT6A-2 or si-CCT6A-3 decreased the confluence of cultured U-2 OS cells. Cell counting using Trypan blue staining demonstrated that the transfection of si-CCT6A-2 and si-CCT6A-3 led to significantly lower cell survival rates than those seen in si-Ctrl cells (Fig 2C) ($p <0.05$),

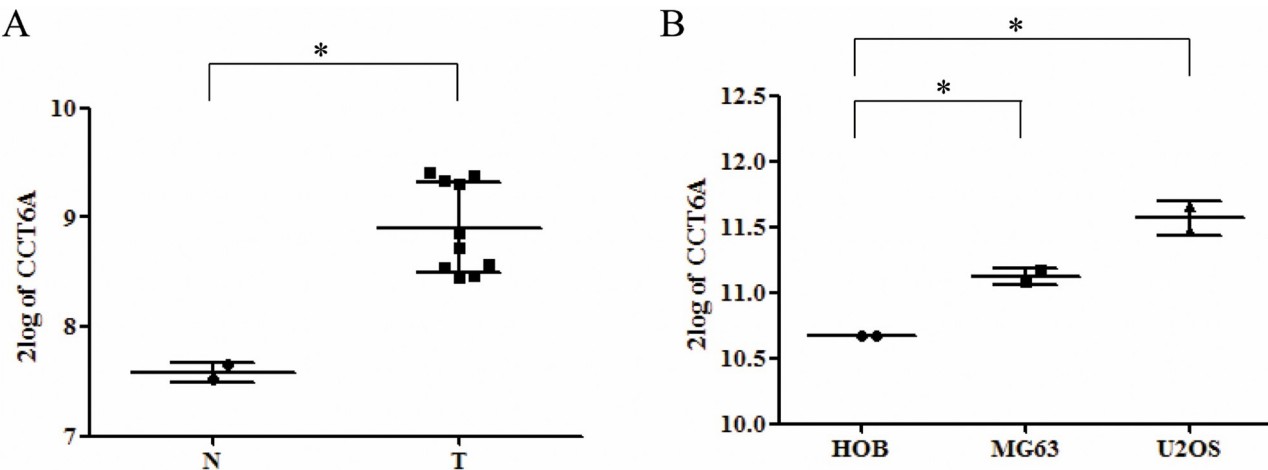

**Fig 1. Expression levels of CCT6A in osteosarcoma tissue and cells.** The R2 genomic analysis and visualization platform were used to analyze the mRNA expression of the protein-coding gene chaperonin-containing TCP1 subunit 6A (CCT6A) in osteosarcoma tissue and cells *$p < 0.05$. mRNA, messenger ribonucleic acid; N, normal tissue; T, cancer tissue.

which further confirmed the suppression of CCT6A knockdown on cell growth in cultured U-2 OS cells.

## CCT6A knockdown inhibits U-2 OS cell proliferation

A CCK8 assay revealed that the cell viability of cultured U-2 OS cells had significantly lower cell viability after transfection with si-CCT6A-2 or si-CCT6A-3 for 48 h or 72 h than si-Ctrl cells ($p < 0.05$) (Fig 3A). Due to its prominent suppression of CCT6A protein expression and cell growth, si-CCT6A-3 was selected for further evaluation as si-CCT6A. Colony formation analysis showed that CCT6A knockdown reduced the formation of colonies from that seen in the si-Ctrl cells ($p < 0.05$) (Fig 3B). Thus, CCT6A knockdown appears to inhibit the proliferation of U-2 OS cells.

## CCT6A knockdown induces U-2 OS cell cycle progression

PI staining and FACS analysis were performed to detect the cell cycle progression of U-2 OS cells after CCT6A knockdown. As shown in Fig 4, the percentage of U-2 OS cells transfected with si-CCT6A (si-CCT6A-3) in the G0/G1 phases was significantly greater (80.49[±2.38]%) than the percentage in si-Ctrl cells (70.53[±0.53]%) ($p < 0.05$). The percentage of cells in the S phase was significantly lower ($p < 0.05$; 8.73[±1.85]% in si-CCT6A cells than in si-Ctrl cells (16.52[±0.40]%) ($p < 0.05$), demonstrating that CCT6A knockdown induces cell cycle arrest in the G0/G1 phases. This might contribute to the suppression of cell proliferation in these cells.

## CCT6A knockdown induces U-2 OS cell apoptosis

Annexin V staining followed by FACS analysis showed a significant right shift in the fluorescence peak and a significantly higher percentage of annexin V positively stained U-2 OS cells after CCT6A knockdown than in si-Ctrl cells (59.93[±3.38]%), or untreated U-2 OS cells (Fig 5) (4.50[±0.71]%) ($p < 0.05$). This suggests that CCT6A plays a critical role in OS cell apoptosis.

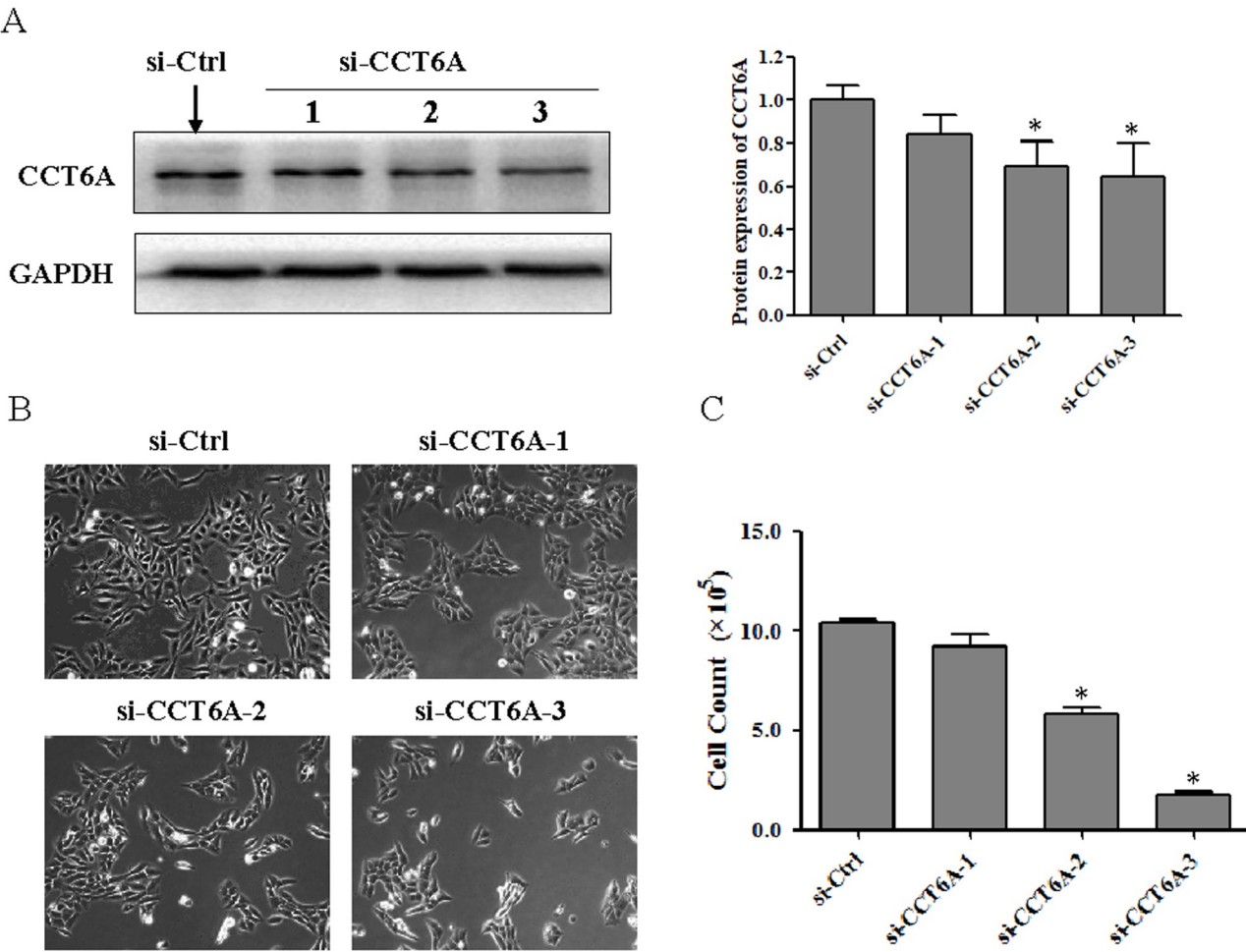

**Fig 2. The effect of CCT6A knockdown on U-2 osteosarcoma cell growth.** U-2 osteosarcoma cells were transfected with three different si-CCT6As or si-Ctrls for 72 h. (A) and Western blot analysis was performed to determine the protein expression of CCT6A. (B) The cells were counted with a Countstar Automated Cell Counter using the Trypan blue exclusion principle. (C) Morphological changes were observed by phase-contrast microscopy at ×100 magnification. The images shown represent three independent experiments. $^*p < 0.05$ vs. si-Ctrl. CCT6A, chaperonin-containing TCP1 subunit 6A; siCT6A-3, CCT6A siRNA; si-Ctrl, control siRNA; siRNA, small interfering ribonucleic acid.

## CCT6A knockdown upregulates Bax and downregulates p-Akt, Bcl-2, and cyclinD1 protein expression

We performed a Western blot analysis to explore the underlying mechanism of CCT6A knockdown on the suppression of U-2 OS cell growth. This showed significantly lower expression of p-Akt because of CCT6A knockdown but had only minor effects on total Akt expression. Further comparison of downstream Akt activation in U-2 OS cells and si-Ctrl cells found that CCT6A knockdown increased the expression of pro-apoptotic protein Bax and decreased the protein expression of anti-apoptotic Bcl-2 and cell cycle regulator cyclinD1 ($p < 0.05$) but had no significant effect on CDK4 expression (Fig 6). These findings suggest that CCT6A knockdown suppresses cell growth through the suppression of Akt pathway activity.

## Discussion

There is an accumulating body of research evidence indicating that the dysregulation of multiple genes contributes to the development and progression of OS [29–31]. In the current study,

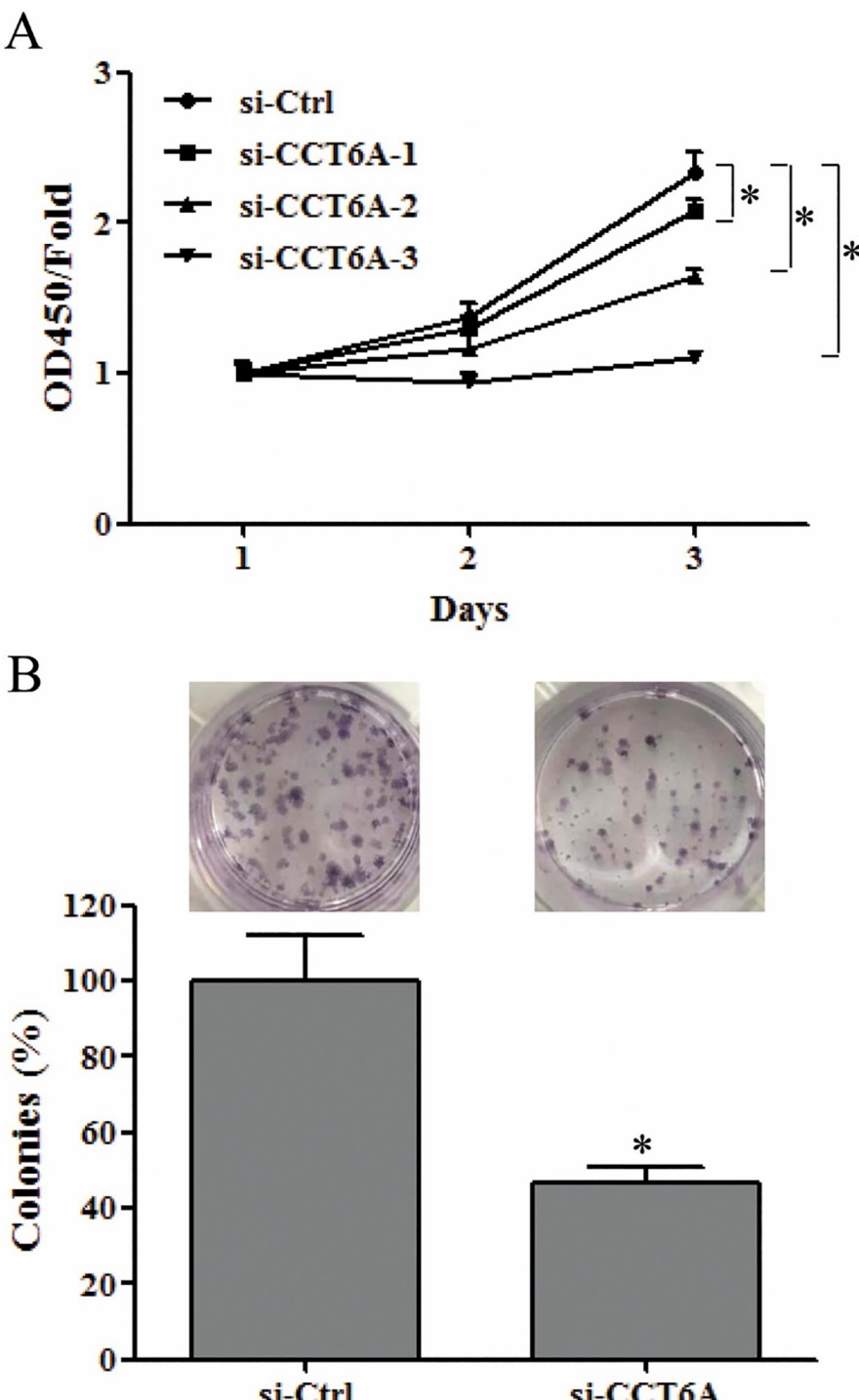

**Fig 3. The effect of CCT6A on U-2 osteosarcoma cell colony formation.** (A). CCK-8 was used to detect the cell viability of U-2 OS cells transfected with three different si-CCT6A oligonucleotides or si-Ctrl for 24 h, 48 h, or 72 h. Data were normalized to viability on day 1 and shown as fold changes. (B) Cell survival was determined by colony formation analysis of the U-2 OS cells transfected with si-CCT6A-3 for 72 h and the si-Ctrl. The images shown represent three independent experiments and the quantification of our colony formation analysis. Data were normalized to the survival of the control cells and are shown as the mean (±SD) (error bars) from three independent experiments. *$p < 0.05$ vs. si-Ctrl. CCK-8, cell-counting kit-8; CCT6A, chaperonin-containing TCP1 subunit 6A; OS, osteosarcoma; SD, standard deviation; siCT6A-3, CCT6A siRNA; si-Ctrl, control siRNA; siRNA, small interfering ribonucleic acid.

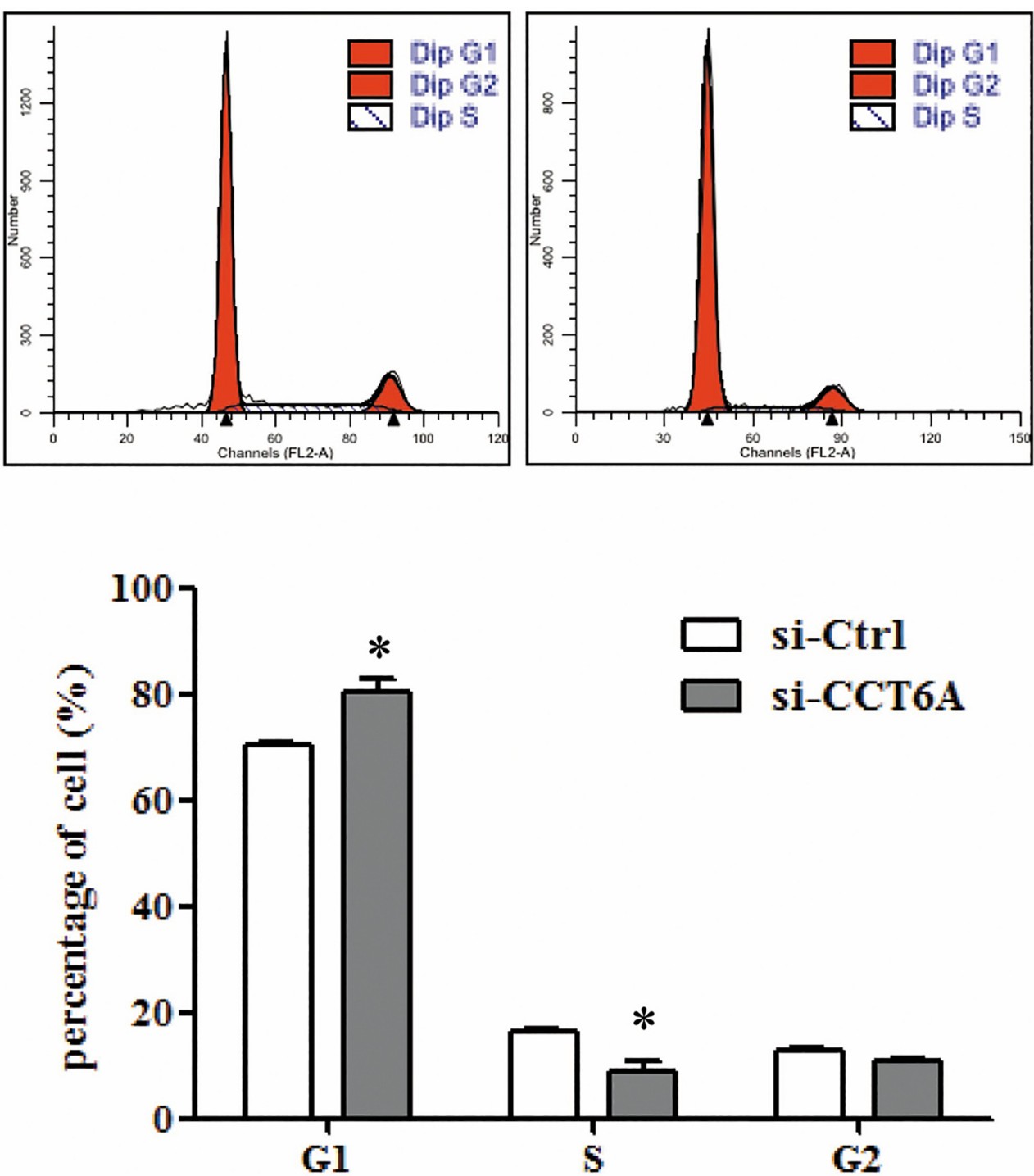

**Fig 4. The effect of CCT6A on U-2 osteosarcoma cell cycle progression.** After transfection with si-Ctrl or si-CCT6A-3 for 72 h, U-2 osteosarcoma cells were stained with PI solution, and cell cycle distribution was analyzed using flow cytometry. The percentages of cells in the G0/G1, S, and G2/M phases were counted. Data are presented as mean (±SD). n = 3. *$p < 0.05$ vs. si-Ctrl. CCT6A, chaperonin-containing TCP1 subunit 6A; PI, propidium iodide; SD, standard deviation; siCT6A-3, CCT6A siRNA; si-Ctrl, control siRNA; siRNA, small interfering ribonucleic acid.

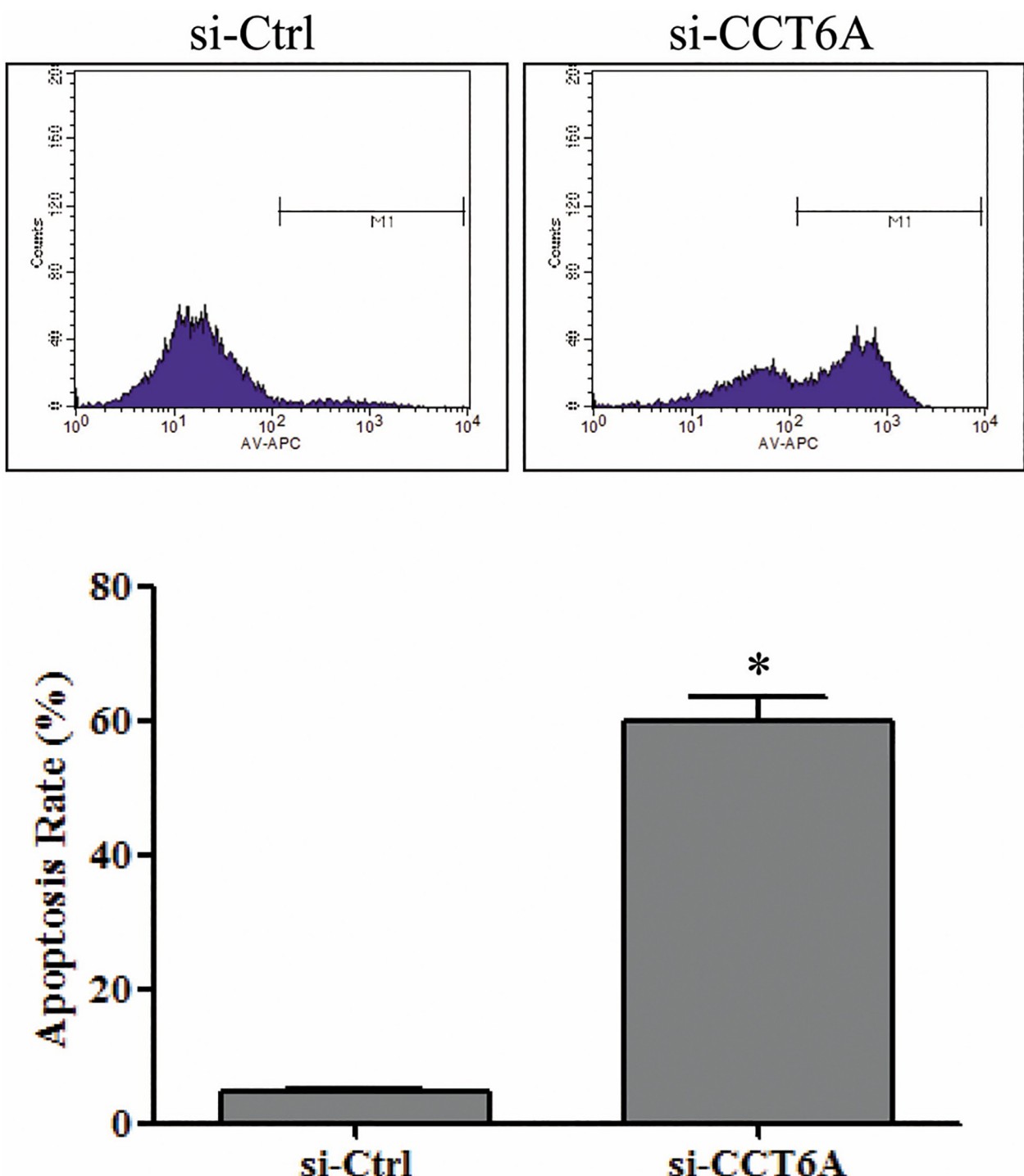

**Fig 5. The effect of CCT6A on U-2 osteosarcoma cell apoptosis.** After transfection with si-CCT6A-3 or control siRNA (si-Ctrl) for 72 h, U-2 osteosarcoma cells were stained with APC annexin V solution, and cell apoptosis was analyzed using flow cytometry. Data are presented as mean (±SD). n = 3. *$p < 0.05$ vs. si-Ctrl. APC, allophycocyanin; CCT6A, chaperonin-containing TCP1 subunit 6A; SD, standard deviation; siCT6A-3, CCT6A siRNA; si-Ctrl, control siRNA; siRNA, small interfering ribonucleic acid.

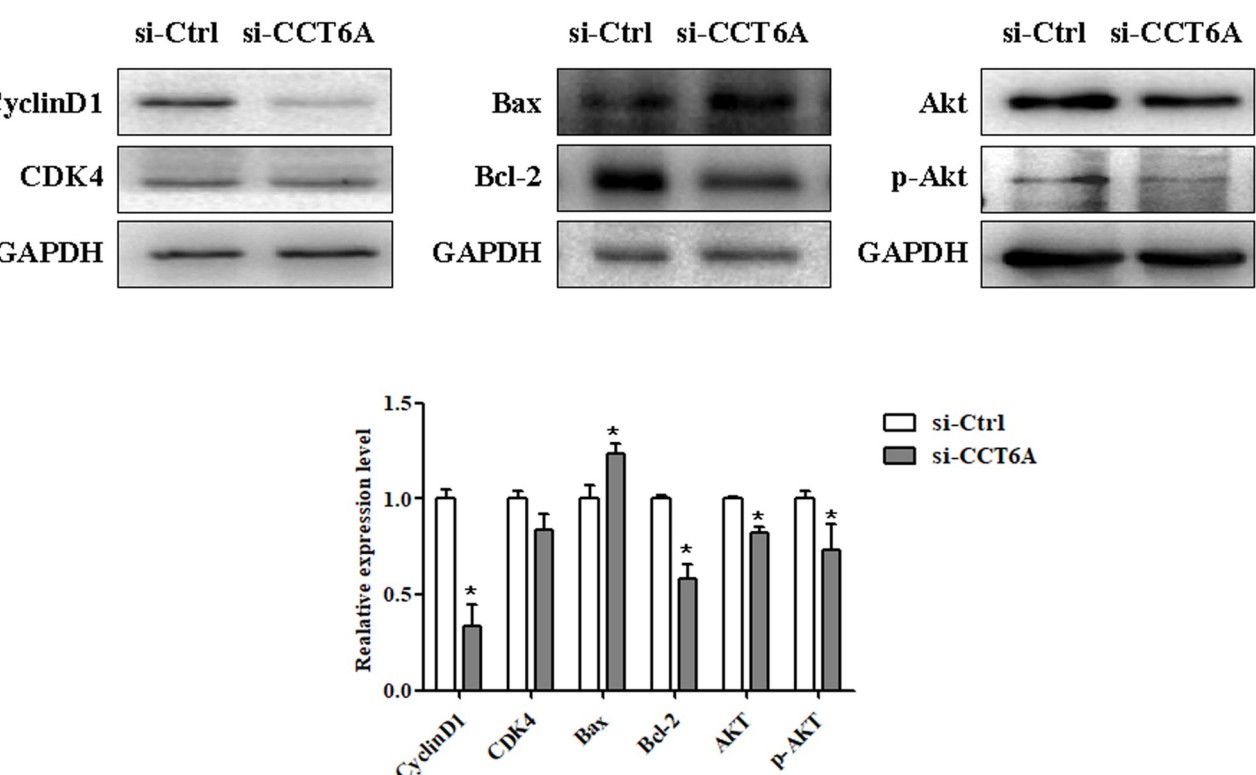

**Fig 6. The effect of CCT6A on activation pathways and its downstream expression in U-2 osteosarcoma cells.** After transfection with si-Ctrl or si-CCT6A-3 for 72 h, the protein expression of p-Akt, Akt, Bax, Bcl-2, cyclinD1, and CDK4 were determined by Western blot analysis. Representative images are shown in the upper panel and the quantification of their expression using ImageLab software is shown in the lower panel. GAPDH was used as an internal control. *$p < 0.05$ vs. si-Ctrl. Akt, protein kinase B; Bax, Bcl-2-associated X-protein; Bcl-2, B-cell lymphoma-2; CCT6A, chaperonin-containing TCP1 subunit 6A; CDK4, cyclin-dependent kinase 4; GAPDH, glyceraldehyde 3-phosphate dehydrogenase; p-Akt, phospho-protein kinase B; si-CCT6A, chaperonin-containing TCP1 subunit 6A; si-Ctrl, control small interfering ribonucleic acid; siRNA, small interfering ribonucleic acid.

analysis of CCT6A mRNA expression in several datasets using the R2 platform [26–28] showed significantly increased CCT6A expression in OS tissue and cells compared to that in non-neoplastic primary osteoblasts. Functional studies revealed that CCT6A knockdown by transfection of CCT6A-specific siRNA suppressed cell growth and decreased the number of OS cells. Moreover, CCT6A knockdown significantly reduced cell viability and colony formation. We also found that CCT6A knockdown induced cell apoptosis and cell cycle arrest at G0/G1. Western blotting revealed that CCT6A knockdown significantly increased Bax and down-regulated p-Akt, Bcl-2, and cyclinD1 protein expression. It had minor effects on Akt protein expression but no effect on CDK4 protein expression. These findings suggest that the increased CCT6A expression in OS cells plays a critical role in OS cell proliferation and apoptosis through the activation of the Akt pathway.

CCT had been reported to play an essential role in the cell cycle [8], tumor growth [9–11], and migration in cancer cells [12, 13]. However, the clinical significance, biological function, and underlying mechanism of the eight subunits of a chaperonin in various cancers, including OS, remain largely unknown. Recent studies indicate that CCT6A expression is significantly increased in ten human tumor cell lines and drug-resistant human melanoma cell lines [23, 24]. Therefore, we analyzed the expression of CCT6A in OS tissue and cells using the R2 application [26–28] and found that CCT6A expression was significantly higher in OS tissue compared to non-neoplastic primary osteoblasts. In addition to an increase in the levels of CCT subunits in cancer cells [32], we found CCT6A mRNA expression to be greater in OS cells than in normal

human osteoblasts. These results suggest that increased CCT6A expression might be common during the development of OS and may play an essential role in OS cell growth regulation. This encouraged us to further explore the role of CCT6A in OS. However, protein expression and the clinical significance of CCT6A in OS should be further addressed in future research.

There is evidence to indicate that CCT is required for the production of native actin and tubulin proteins, and therefore participates in cell cycle progression [8]. As a subunit of the CCT family, CCT6A is essential for the correct folding and oligomerization or polymerization of native proteins, with actin and tubulin being the major substrates [17]. Therefore, we assessed the role of CCT6A in OS cell proliferation. As expected, CCT6A knockdown reduced cell confluence and the number of cultured OS cells, indicating that CCT6A knockdown suppresses OS cell growth. We further revealed that CCT6A knockdown decreased cell viability and colony formation. Consistent with a previous functional study of CCT [8], we found that CCT6A knockdown increased cell cycle arrest in the G0/G1 phase and downregulated the expression of cyclinD1 in cultured OS cells, suggesting that the downregulation of cyclinD1--mediated cell cycle arrest contributes to the suppression of cell proliferation.

Due to the essential role of CCT in tumor growth [9–11], we further evaluated the effect of CCT6A knockdown on OS cell apoptosis. As in previous studies, CCT6A knockdown was found to increase OS cell apoptosis. Previous research has shown that increased levels of the anti-apoptotic protein Bcl-2 and decreased levels of the pro-apoptotic protein Bax contribute to cell apoptosis [33, 34]. Therefore, we explored the underlying mechanism of CCT6A knockdown further through the determination of Bcl-2 and Bax expression. We found that CCT6A knockdown significantly downregulates Bcl-2 expression while increasing Bax expression. However, the mechanism of CCT6A knockdown on the induction of cell apoptosis needs to be addressed by further research.

Previous OS studies indicate that hyperactivation of the Akt pathway plays a critical role in tumorigenesis, proliferation, cycles, and apoptosis in OS cells by regulating the downstream expression of cyclinD1, CDK4, Bcl2, and Bax [35–37]. Therefore, we detected this activation and found that, while CCT6A knockdown decreased the expression of p-Akt, it had only minor effects on the total expression of Akt. this suggests that suppression of the Akt pathway might be among the underlying mechanisms behind the suppression of OS cell growth by CCT6A knockdown. However, the regulatory effects of CCT6A on other signaling pathways (including the signal transducer and activator of transcription 3 [STAT3] and mitogen-activated protein kinase [MAPK] pathways) should be addressed in future studies. Our study has some potential limitations, because we focused on the role of CCT6A in osteosarcoma cells without considering its effect on the tumor microenvironment and immune components. Further study, we will investigate that CCT6A participates in both microenvironment and adaptive immune responses in OS cells.

## Conclusions

The present study has revealed increased CCT6A mRNA expression in OS tissues and cells and demonstrated that CCT6A knockdown suppresses OS cell growth by inhibiting cell proliferation and inducing cell apoptosis through suppression of Akt pathway activation in OS cells. These findings suggest that CCT6A might be an effective target for anticancer treatment.

## Supporting information

**S1 Data.**
(PDF)

**S1 File.**
(PDF)

## Acknowledgments

The authors are grateful to all participating patients and their families.

## Author Contributions

**Data curation:** Weiquan Zeng, Lihui Wei, Yi Fang, Jun Peng, Aling Shen.

**Formal analysis:** Weiquan Zeng, Lihui Wei, Yi Fang, Youqin Chen.

**Funding acquisition:** Aling Shen.

**Investigation:** Meizhu Wu, Liya Liu, Qiurong Xie, Youqin Chen.

**Methodology:** Weiquan Zeng, Meizhu Wu, Ying Cheng, Liya Liu, Yuying Han, Qiurong Xie, Lihui Wei, Yi Fang, Youqin Chen.

**Resources:** Yuying Han.

**Software:** Ying Cheng, Liya Liu, Jiapeng Li.

**Supervision:** Jun Peng.

**Validation:** Jiapeng Li.

**Visualization:** Jiapeng Li.

**Writing – original draft:** Weiquan Zeng, Meizhu Wu, Ying Cheng.

**Writing – review & editing:** Jun Peng, Aling Shen.

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
