## [Decision Letter · Decision Letter 0]

19 Oct 2022

PONE-D-22-26167CCT6A knockdown suppresses osteosarcoma cells growth and AKT pathway activation in vitroPLOS ONE

Dear Dr. Shen

Thank you for submitting your manuscript to PLOS ONE. After careful consideration, we feel that it has merit but does not fully meet PLOS ONE’s publication criteria as it currently stands. Therefore, we invite you to submit a revised version of the manuscript that addresses the points raised during the review process.

We look forward to receiving your revised manuscript.

Kind regards,

Filomena de Nigris, Ph.D.

Academic Editor

PLOS ONE

Journal Requirements:

"This study was supported by National Natural Science Foundation of China (81803882), and Natural Science Foundation of Fujian Province (2020J06026)."

"1、National Natural Science Foundation of China (81803882) . Funders:AS. He is Corresponding author. https://www.nsfc.gov.cn

2、 Natural Science Foundation of Fujian Province (2020J06026). Funders:AS. He is Corresponding author. " ext-link-type="uri" xlink:type="simple">http://xmgl.kjt.fujian.gov.cn/showLoginPage.do?type=fujianloginflag=false"

Reviewers' comments:

Reviewer's Responses to Questions

**Comments to the Author**

1. Is the manuscript technically sound, and do the data support the conclusions?

Reviewer #1: Partly

Reviewer #2: Partly

2. Has the statistical analysis been performed appropriately and rigorously? 

Reviewer #1: Yes

Reviewer #2: N/A

3. Have the authors made all data underlying the findings in their manuscript fully available?

Reviewer #1: No

Reviewer #2: No

4. Is the manuscript presented in an intelligible fashion and written in standard English?

Reviewer #1: No

Reviewer #2: No

5. Review Comments to the Author

Reviewer #1: The data do not support all the findings fully. Several additional experiments need.

1 in the abstract section there are several grammatica errors and repetitions

2 R2 wich kind of database is? Only two normal tissue and ten tumors were analyzed?

3 How the authors giustificate a different effects of different clones if they express the same amount of residual CCT6A protein?

4 Wich clones was selected? Plaese indícate in figure 3 and in figure 4.

5 In figure 5 the authors evaluated apoptosys by facs. How they say the apoptosys rate was sixty %?

6 Counts on y axe is not a % of cells.

7 Could the authors explain the pleiotropic role of CCT6A in transcriptional regiulation of several genes.

Discussion

In discussion section, please at line 245 also consider the effects of microenviroment in osteosarcoma progression in particular of immune components. See Mosca et all 2022.

Reviewer #2: In the abstract section there are several grammatical errors and repetitions

2. From Figure 1 it is clear that 2 samples from normal tissues and 10 from OSTEOSARCOMA tissues were analyzed, a number that appears quite small

3.The images are not immediately understandable because they are not completely detailed, in order to be able to interpret them well you need a careful reading of the text. For example in the figure 2 it is not reported that the test concerns the U2OS cell line, which is reported in the text (lines 197-199); Figure 2B is described only in lines 513 and 514 (Legend of the figures) in which it is said that the figure represents the number of cells counted using Countstar Automated Cell Counter, method described in paragraph 2.5 (lines 133 to 137).

4. It would have been useful to obtain more information on viability by setting up a flow cytometric assay that simultaneously assessed both the positivity to the staining with ANNESSIN V and the permeability to PI so as to highlight the necrotic cells (ANNESSINA V- and PI +) and those that were late apoptotic (ANNESSINA V + and PI + )

5.Regarding the Analysis of cellular colony formation: it would be useful to understand if the image shown in figure 3B is still affected by the transfection or if it is reasonable to think that the vector has been eliminated from the cells after transfection (72 hours)

6. PLOS authors have the option to publish the peer review history of their article (what does this mean?). If published, this will include your full peer review and any attached files.

Reviewer #1: No

Reviewer #2: No

---

## [Author Response · Author response to Decision Letter 0]

2 Dec 2022

Dear Dr Nigris and reviewers, 

Great appreciate for all your valuable comments. We make a point-to-point response as following: 

Thanks. We revised the manuscript base on the templates. Please find the revised manuscript.

"This study was supported by National Natural Science Foundation of China (81803882), and Natural Science Foundation of Fujian Province (2020J06026)."

"1、National Natural Science Foundation of China (81803882) . Funders:AS. He is Corresponding author. https://www.nsfc.gov.cn

2、 Natural Science Foundation of Fujian Province (2020J06026). Funders:AS. He is Corresponding author. http://xmgl.kjt.fujian.gov.cn/showLoginPage.do?type=fujianloginflag=false"

Sure. We have removed the funding information from the Acknowledgments Section and updated in the Funding Statement, as well as stated the funding information in the cover letter 

3. PLOS ONE now requires that authors provide the original uncropped and unadjusted images underlying all blot or gel results reported in a submission’s figures or Supporting Information files. This policy and the journal’s other requirements for blot/gel reporting and figure preparation are described in detail at https://journals.plos.org/plosone/s/figures#loc-blot-and-gel-reporting-requirements and https://journals.plos.org/plosone/s/figures#loc-preparing-figures-from-image-files.

Sure, we have added the PDF about all blot picture, please find the attachment.

1. Is the manuscript technically sound, and do the data support the conclusions?

All the data have three times repeat or six times repeat (CCK-8 assay), which were stated in figure legends sections within the revised manuscript. All data are represented as mean and SD. It makes sure the conclusions have drawn appropriately based on the data presented. 

2. Has the statistical analysis been performed appropriately and rigorously? 

All the statistical analyses were conducted using SPSS statistical software (version: 22.0). Data were presented as the mean ± standard deviation of at least three independent experiments. The difference between two groups were tested using the independent Student's t test, and among or more than 3 groups were analyzed using one-way ANOVA. P 0.05 was considered significant.

3. Have the authors made all data underlying the findings in their manuscript fully available?

Sure, we provided the original data, including western blot picture and others, please find the attachment.

4. Is the manuscript presented in an intelligible fashion and written in standard English?

Thanks. The manuscript has been carefully reviewed by an experienced editor whose first language is English and who specializes in editing papers written by scientists whose native language is not English. Please find the attachment about CERTIFICATE OF EDITING.

Reviewer #1: The data do not support all the findings fully. Several additional experiments need.

1 in the abstract section there are several grammatica errors and repetitions

Thanks. Same as above, the manuscript has been carefully reviewed by an experienced editor whose first language is English and who specializes in editing papers written by scientists whose native language is not English. Please find the attachment about CERTIFICATE OF EDITING.

2 R2 wich kind of database is? Only two normal tissue and ten tumors were analyzed?

R2 Genomics Analysis and Visualization Platform, is an online datamining and discovery platform designed to assist the bio-medical researchers with limited to no Bioinformatics skills to perform data science tasks in the omics field. After double check, we only found two normal tissue and ten tumors in the R2 website. Due to the limited number of both normal and tumor tissues, we will collect the more samples for verification of CCT6A expression in our recently study.

3 How the authors giustificate a different effects of different clones if they express the same amount of residual CCT6A protein?

In Western-blot assay, BCA assay was performed according to the manufacturer's instructions to detect the total protein concentration. 50 μg of the total protein (all of samples with same protein concentration) was loaded and separated. Moreover, we also calculated the quantification of the protein expression using ImageLab software which supporting the different expression of CCT6A. We could make sure the protein expression of CCT6A in the same protein concentration. 

4 Wich clones was selected? Plaese indícate in figure 3 and in figure 4.

Due to its prominent suppression of CCT6A protein expression and cell growth, si-CCT6A-3 was selected for further evaluation as si-CCT6A, which were indicated the figure legends section. 

5 In figure 5 the authors evaluated apoptosys by facs. How they say the apoptosys rate was sixty %?

The cells in M1 phase were recognized the total Annexin V positive staining (about 60% in si-CCT6A group).

6 Counts on y axe is not a % of cells.

Thank you very much. We calculated the percentage of cells in different phases based on the counts in the export data from FACS. So we presented the percentage of cells on y axe.

7 Could the authors explain the pleiotropic role of CCT6A in transcriptional regiulation of several genes.

Our results shown that CCT6A knockdown decreased the expression of phosphorylation of Akt, it had only minor effects on the total expression of Akt. This suggests that suppression of the Akt phosphorylation might be among the underlying mechanisms behind the suppression of OS cell growth and the induction of apoptosis by CCT6A knockdown.

Discussion

In discussion section, please at line 245 also consider the effects of microenviroment in osteosarcoma progression in particular of immune components. See Mosca et all 2022.

Our study has some potential limitations, because we focused on the role of CCT6A in osteosarcoma cells without considering its effect on the tumor microenvironment and immune components. Further study, we will investigate that CCT6A participates

in both microenvironment and adaptive immune responses in OS cells.

---

## [Decision Letter · Decision Letter 1]

15 Dec 2022

CCT6A knockdown suppresses osteosarcoma cell growth and Akt pathway activation in vitro

PONE-D-22-26167R1

Dear Dr. Shen

We’re pleased to inform you that your manuscript has been judged scientifically suitable for publication and will be formally accepted for publication once it meets all outstanding technical requirements.

Kind regards,

Filomena de Nigris, Ph.D.

Academic Editor

PLOS ONE

Additional Editor Comments (optional):

Reviewers' comments:

Reviewer's Responses to Questions

**Comments to the Author**

1. If the authors have adequately addressed your comments raised in a previous round of review and you feel that this manuscript is now acceptable for publication, you may indicate that here to bypass the “Comments to the Author” section, enter your conflict of interest statement in the “Confidential to Editor” section, and submit your "Accept" recommendation.

Reviewer #1: All comments have been addressed

Reviewer #2: All comments have been addressed

2. Is the manuscript technically sound, and do the data support the conclusions?

Reviewer #1: Yes

Reviewer #2: Yes

3. Has the statistical analysis been performed appropriately and rigorously? 

Reviewer #1: Yes

Reviewer #2: Yes

4. Have the authors made all data underlying the findings in their manuscript fully available?

Reviewer #1: No

Reviewer #2: Yes

5. Is the manuscript presented in an intelligible fashion and written in standard English?

Reviewer #1: Yes

Reviewer #2: Yes

6. Review Comments to the Author

Reviewer #1: the comments are satisfactory. I don’t have other questions about the manuscript. I understand the explanations

Reviewer #2: The authors responded sufficiently to the comments previously made and made the data underlying the findings fully available in their manuscript

7. PLOS authors have the option to publish the peer review history of their article (what does this mean?). If published, this will include your full peer review and any attached files.

Reviewer #1: No

Reviewer #2: No

---

## [Editor Report · Acceptance letter]

21 Dec 2022

PONE-D-22-26167R1 

CCT6A knockdown suppresses osteosarcoma cell growth and Akt pathway activation *in vitro*

Dear Dr. Shen:

I'm pleased to inform you that your manuscript has been deemed suitable for publication in PLOS ONE. Congratulations! Your manuscript is now with our production department. 

Kind regards, 

on behalf of

Prof. Filomena de Nigris 

Academic Editor

PLOS ONE